# Single-cell methylation analysis of brain tissue prioritizes mutations that alter transcription

## Graphical abstract

Single nucleus methylation sequence profiles from hippocampus of eight inbred mice from uniform manifold approximation and projection (UMAP)

At densities less than 40 CpG/Kb there are more mutations in methylated than unmethylated DNA

In regions with a high density of cytosine/guanine dinucleotides ( CpG ), mutations are constrained to preserve the same sites of methylation. In regions with low CpG density there is less constraint

The impact of mutations on transcript abundance depends on CpG density.

## Authors

Jonathan Flint, Matthew G. Heffel, Zeyuan Chen, ..., Patrick B. Chen, Jason Ernst, Chongyuan Luo

## Correspondence

jflint@mednet.ucla.edu

## In brief

Flint et al. examine the relationship between sequence variation and methylation in the brains of inbred mouse strains. They find that mutations that alter methylation influence transcript abundance dependent on local CpG sequence density. The effect is larger in regions of high density and lower in regions of low density.

## Highlights

- Density of CpG sequence is not linearly correlated with methylation rates

- Regions of low- and high-density CpG sites differ in their methylation conservation

- DNA mutations in high, but not low, CpG density regions affect RNA abundance

- Mutations contributing to RNA abundance are enriched in active enhancers

Flint et al., 2023, Cell Genomics 3, 100454
December 13, 2023 © 2023 The Author(s).

CellPress

## Article

# Single-cell methylation analysis of brain tissue prioritizes mutations that alter transcription

Jonathan Flint,[1,2,5,*] Matthew G. Heffel,[2] Zeyuan Chen,[3] Joel Mefford,[2] Emilie Marcus,[4] Patrick B. Chen,[1] Jason Ernst,[3,4] and Chongyuan Luo[2]

[1]Department of Psychiatry and Biobehavioral Sciences, University of California Los Angeles, Los Angeles, CA, USA
[2]Department of Human Genetics, David Geffen School of Medicine, University of California Los Angeles, Los Angeles, CA, USA
[3]Department of Computer Science, Samueli School of Engineering, University of California Los Angeles, Los Angeles, CA, USA
[4]Department of Biological Chemistry, David Geffen School of Medicine, University of California Los Angeles, Los Angeles, CA, USA
[5]Lead contact
*Correspondence: jflint@mednet.ucla.edu

## SUMMARY

Relating genetic variants to behavior remains a fundamental challenge. To assess the utility of DNA methylation marks in discovering causative variants, we examined their relationship to genetic variation by generating single-nucleus methylomes from the hippocampus of eight inbred mouse strains. At CpG sequence densities under 40 CpG/Kb, cells compensate for loss of methylated sites by methylating additional sites to maintain methylation levels. At higher CpG sequence densities, the exact location of a methylated site becomes more important, suggesting that variants affecting methylation will have a greater effect when occurring in higher CpG densities than in lower. We found this to be true for a variant's effect on transcript abundance, indicating that candidate variants can be prioritized based on CpG sequence density. Our findings imply that DNA methylation influences the likelihood that mutations occur at specific sites in the genome, supporting the view that the distribution of mutations is not random.

## INTRODUCTION

Heritable effects on behavior in inbred mice are pervasive, frequently large, and thought to be associated with heritable differences in neuronal composition and neuroanatomy.[1–10] Yet, despite hundreds of genetic mapping studies,[11] access to nearly complete sequences of multiple strains,[12,13] and catalogs of cell types and their respective genomic properties,[14–17] scant progress has been made toward relating genetic variants to behavior.

One impediment to progress is that due to the relatively large intervals into which quantitative trait loci are mapped, where there are usually thousands, and often tens of thousands, of candidate causal variants. How are these variants to be prioritized for functional study, and is there another level in addition to genetic variation that determines their function? The majority of causative variants lie in regulatory regions of the genome and likely act by altering a molecular phenotype,[18–20] such as DNA methylation, which is associated with neuronal function and behavior.[21]

DNA methylation occurs at regulatory elements in the genome, affecting transcription factor binding affinity and controlling gene transcription,[22,23] roles that suggest it may play a role in mediating the effect of sequence variants that alter behavior. While both CpG and CH methylation (mCG and mCH, where H = A, C, or T) show cell-type specificity,[22–24] only methylation at CpG dinucleotides propagates through cell division, providing stable marks that differentiate cell types.[16] Methylation's cell-type specificity in the brain means that mutations, as much as they act through methylation, will have different consequences depending on the cell type that they affect. Hence single-cell data from multiple individuals are needed to understand the relationship between mutation, methylation state, and behavioral outcome. However, to use CpG methylation marks to discover causative variants requires an understanding of the relationship between sequence and methylation variation.[25]

What happens when a mutation removes a methylated CpG site? Broadly speaking, there are three possible consequences. The simplest, at least for interpreting genetic association studies, is that the mutation disrupts a sequence motif, with consequences for whatever function that motif performs. For example, methylated CpGs can directly inhibit or augment transcription factor binding,[26] and a mutation could interfere with this process.[27–29] Knowing the location of the mutation at a methylated site within a known motif or regulatory region will suggest candidate proteins. Second, the function of methylation may depend on the density of methylated cytosines in a region. For instance, CpG density is associated with active histone marks and high expression,[30–32] and CpGs contribute to transcriptional activity regardless of whether they are part of a sequence motif.[33] In this case, the consequences of a mutation will depend on local context (e.g., how many other cytosines are methylated), rather than on the specific sequence. Third, there is the possibility that methylation function and DNA sequence are independent. If identical DNA sequences are differentially methylated,

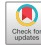

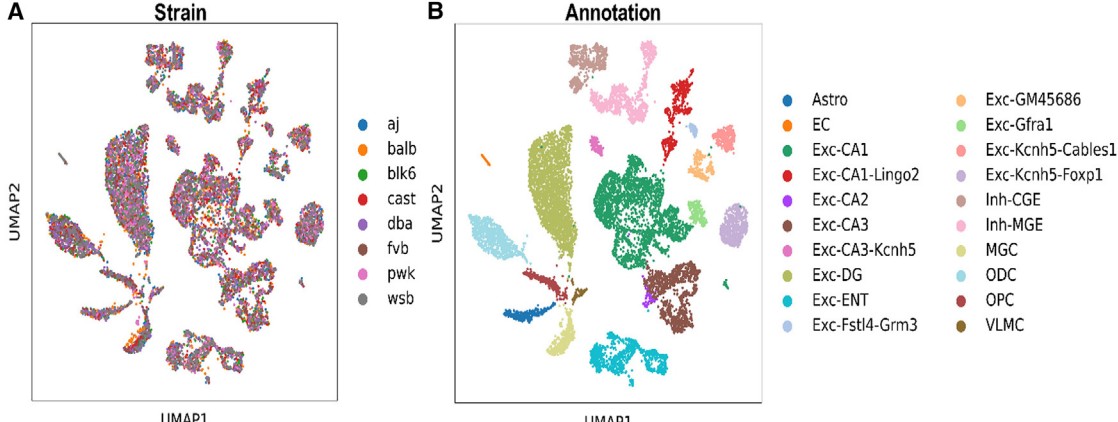

**Figure 1. UMAP of cell-type clusters in ventral hippocampus derived from methylation**
Cells are colored by (A) strain of origin and (B) cell-type identity. aj = A/J, b6 = C57BL/6J, balb = BALB/cJ, cast = CAST/EiJ, d2 = DBA/2J, fvb = FVB/J, pwk = PWK/PhJ, and wsb = WSB/EiJ.

methylation could be a mechanism through which external experiences alter gene function, and hence phenotypes, in a stable manner.[34–36]

In this paper, we examine the relationship between methylation and sequence variation using data from eight inbred mouse strains. Existing methylomes are mostly derived from one strain (C57BL/6J, B6)[16,17,37–39] or are from array-based assays that do not interrogate all methylation sites at a single-cell level in the brain.[40] We generated genome-wide, base-resolution maps of multiple cell types from the hippocampus in each of the eight strains. We chose CAST/EiJ, a fully sequenced representative of *M. m. castaneus*, as an outgroup and compared methylation and sequence variation to those of five classical laboratory strains (A/J, C57BL/6J, BALB/cJ, FVB/J, and DBA/2J) (all *M. m. domesticus*) and two wild-derived inbred strains: WSB/EiJ (*M. m. domesticus*) and PWK/PhJ (*M. m. musculus*). We show that interpretation of the functional consequences of sequence variation, as mediated by methylation, depends on local CpG density.

## RESULTS

### Cell clusters identified from methylation profiles in eight mouse strains

We generated 13,683 single-nucleus methylation sequence (snmC-seq) profiles from microdissected ventral hippocampus tissue of eight mouse strains (Figure 1). We chose the ventral hippocampus because the cytoarchitecture of the hippocampus is less complex than other parts of the brain, and because many behaviors that are involved in hippocampal function have been mapped.[11] 11,694 of these profiles passed our quality control (QC) metrics. Strains were sampled in duplicate, and fluorescence-activated nuclei sorting was used to isolate approximately 83% NeuN-positive and 17% NeuN-negative cells to enrich for neurons.

We performed iterative clustering analysis on the mC dataset based on similarity of global CG and CH methylation in 100-kb bins of the mouse genome. Using this approach, we identified a total of 20 distinct cell types within the ventral hippocampus.

We identified cells belonging to every major subregion of the ventral hippocampus and to every major cell-type class (astrocytes, microglia, excitatory neurons, inhibitory neurons) based on the hypo-methylation states of multiple canonical markers of each cell type. Other neuronal types that were not confidently ascribed to a hippocampal subregion were labeled first by whether they express inhibitory or excitatory markers and then by differences in CH hypomethylation of genes. These ambiguous subregion clusters may relate to other known neuronal subtypes within the hippocampus that were previously defined by RNA sequencing.

Sequence coverage varied widely among the 20 cell types we identified: from less than 10X coverage for Exc-CA2, Exc-Fstl4-Grm3, EC, Exc-CA3-Kcnh5, VLMC, Exc-Gfra1, and Exc-GM45686 to more than 50 for Exc-DG (Table S1). Genome coverage for each cell type is given in the supplemental information. We decided to use the Exc-DG hippocampal cell type for subsequent analyses, because of the high coverage and because we were unable to identify any subregion clusters, indicating homogeneity.

### Conservation of methylated sites between strains depends on CpG sequence density

To investigate the relationship between DNA methylation and sequence variation, we first looked at the association of methylation with the density of CpGs, following earlier reports of its association with gene expression and active histone marks.[30–33] To do this, we estimated the density of CpG dinucleotides in windows of 2 Kb (this size was chosen to capture regions of clustered methylation; results for 1 kb and 500 bp were the same) and explored its relationship with the number of methylated sites.

Figure 2A shows results for the Exc-DG hippocampal cell type. The distribution of methylated sites, with respect to CpG sequence density, consists of three fractions. First, there is a highly methylated fraction at densities of less than 25 CpG/Kb. At densities between 25 and 40 CpG/Kb DNA, segments are partly methylated, while DNA in which less than 20% of CpGs

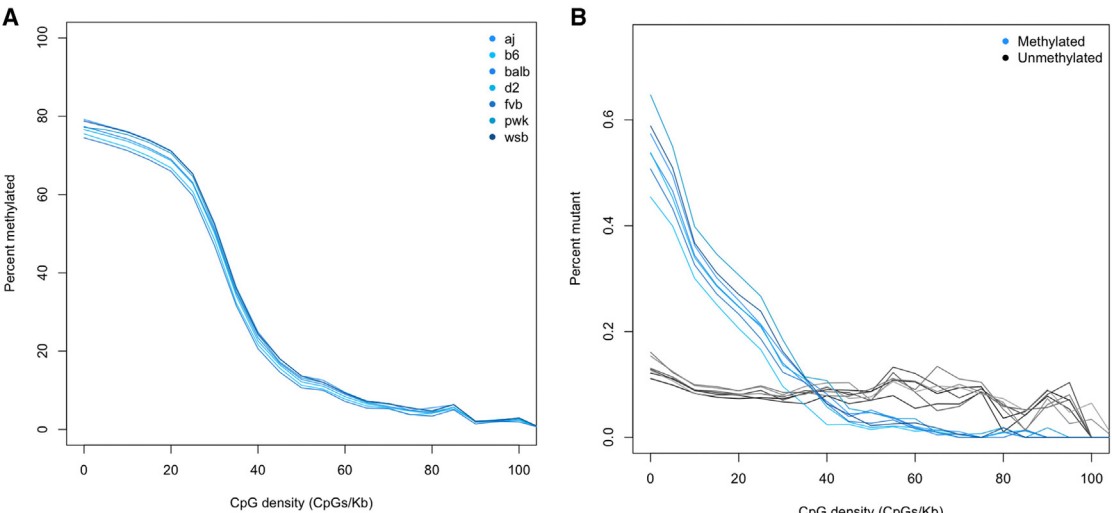

**Figure 2. Relationship between methylation state, mutations, and CpG density**

(A) Percentage of CpG sites that are methylated is shown on the vertical axis. The horizontal axis is the sequence density of CpG sites (regardless of methylation) per kilobase of genomic DNA. Each blue line represents a different inbred strain where aj = A/J, b6 = C57BL/6J, balb = BALB/cJ, d2 = DBA/2J, fvb = FVB/J, pwk = PWK/PhJ, and wsb = WSB/EiJ.

(B) Percentage of CpG sites for the seven inbred strains that are mutated, relative to the outgroup strain CAST/EiJ. The percentages are shown for methylated CpGs (in blue) and non-methylated (in gray). The horizontal axis again shows CpG sequence density per kilobase of genomic DNA.

are methylated occurs in a fraction of high CpG density (greater than 40 CpG/Kb). The same relationship is observed for other cell types for which we have sequence coverage greater than 10-fold. At lower sequence coverages, the estimates of percentage methylation for most of the CpG density segments are too variable to reveal the relationship (illustrated in Figure S2).

We next explored the relationship between CpG sequence density and mutations in the Exc-DG cell type. Using CAST/EiJ as an outgroup, we calculated the percentage of sequence variation (defined as either the loss or the gain of a cytosine in the comparison between the outgroup and the strain; other nucleotides were excluded) in each 2-kb segment for each strain, doing this separately for methylated and non-methylated CpGs. We summed sequence variants in each 2-kb segment and expressed them as a percentage of the total number of CpG sites in the segment.

Figure 2B shows that the percentage of sequence variants in methylated DNA (blue lines) is higher than in unmethylated DNA (gray lines) and that this relationship depends on the density of CpGs. At densities greater than 40 CpG/Kb, relatively more sequence variants are found in unmethylated than methylated DNA, consistent with the observation that mutation of cytosine to thymine at unmethylated CpG dinucleotides has the highest rate of all base substitutions.[41–43] Our results show that CpG sequence density is correlated with both the fraction of methylated CpGs and with the distribution of mutations at methylated CpGs, and that this relationship is non-linear.

## To what extent is the density of methylated CpG determined by sequence?

We turn next to consider what maintains the correlation between two strains in the number of methylated CpG sites in each

segment of DNA. There are two possibilities: the amount of CpG methylation can be the same in two different mouse strains because the same sites are methylated or because the total number of sites is equal in the two strains, regardless of which sites are methylated. To test between these alternatives, we calculated two values for each 2-kb segment of the genome. First, we counted the number of methylated CpGs in each strain. We set the larger number as the denominator to derive a ratio of the two numbers, bounded between 0 and 1. If the ratio is 1, then the segment contains the same number of methylated sites. If the ratio is less than one, it means one strain has fewer methylated sites than the other. Second, we derived a measure of the correlation between sites. We estimated the probability that both sites are in the same state (methylated or not) in each segment. If the probability is 1, this means we can fully predict which sites are methylated in one strain from knowing the state in the other. Probabilities less than 1 mean that knowing the methylation state of sites in one strain is less predictive of their state in the second strain.

Figure 3 shows that for low CpG sequence densities, the ratio is close to 1 for each strain comparison, while the probability of sites being in the same state is about 0.8. In other words, the methylation density is maintained, even though not all homologous sites are methylated in both strains. The pattern changes as densities increase above 40 CpG/Kb, until at densities above 80 CpG/Kb it reverses, with the probability becoming higher than the ratio, though the small sample sizes of hypomethylated CpGs introduce a large variance. Thus, constancy is maintained in two ways: at low density (less than 40 CpG/Kb) different sites may be methylated, while at higher density, it is more likely that identical sites in each strain are methylated.

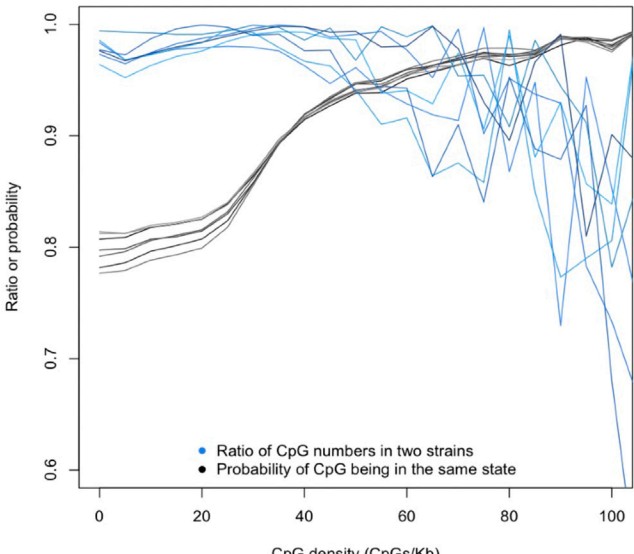

**Figure 3. Comparison of the probability of methylation state with the ratio of the number of methylated sites**

The horizontal axis represents the probability that both sites in two strains are in the same state (methylated or not) in each 2-kb segment (black lines) and also the ratio of the total number of methylated sites in two strains, again for each 2-kb segment (blue lines). Each line is a separate strain comparison, which is the same as that shown in Figure 1B. The horizontal axis shows the density of CpG sites (regardless of methylation) per kilobase of genomic DNA.

## The impact of mutations at methylated CpG sites depends on local CpG sequence density

Our findings suggest that a mutation removing a methylated CpG site may have different phenotypic consequences when it occurs in a region of low compared to high CpG sequence density. Methylation in general is a repressive mark,[25] so we expect that most CpG mutations that disrupt methylation will increase gene expression (by relieving suppression). Consequently, sequence variants disrupting CpG methylated sites in low CpG sequence density regions would be less predictive of transcript abundance than mutations in high CpG sequence density regions. We tested the hypothesis by examining the association between mutations and transcript abundance. Using single-cell RNA data, we compared two strains, B6 and DBA/2J, for the Exc-DG cell type.

We generated single-nuclei transcriptomes for a total of 30,880 nuclei (median UMI counts per cell: 5,816) from two biological replicates that passed our doublet and low-quality filtering steps (STAR Methods). We carried out multi-modal clustering and visualization using uniform manifold approximation and projection (UMAP) to embed transcriptomic and methylation data (Figure 4A), noting that there were no differences by strain (Figure 4B), and that the same major cell-type groups from the methylation dataset were represented, based on the same marker gene set used to annotate the methylation data (Figure 4C). Major cell-type clusters within snRNA and snMethylation were highly consistent between modalities (Figure 4C), except for the small non-neuronal clusters MGC/OPC/VLMC (STAR Methods), which were excluded from further analysis.

We ran DESeq2[44] to identify transcripts that were differentially expressed and, using a liberal threshold of p <0.05 (unadjusted), identified 2,049 transcripts for downstream analysis (we also examined the effect of including transcripts at more conservative thresholds, and we give results in the supplemental information). For mutations, we took all single-nucleotide sites in the mouse genome that altered a methylated CpG present in the strain D2 to either "A" or "T" (to ensure the mutation results in a site that cannot be methylated, regardless of the strand on which it occurs). That yielded 23,959 mutations, of which 1,862 lie within regions with more than 40 CpGs/Kb, categorized as "high" density. We matched the position of each methylated CpG site to a gene interval, defined as running from 2 kb upstream of the transcriptional start site to 2 kb downstream of the 3′ end of the gene.

When mutations occur in high CpG sequence density regions, the mean ratio of D2 to B6 transcripts is significantly higher than it is in low CpG sequence density regions (mean = 0.15 vs. mean = 0.02; t = 4.3, degrees of freedom [df] = 1,924.6, p value = 1.9e−05), supporting the hypothesis that mutations in high CpG sequence density regions are associated with a relatively larger effect. It is possible that some of the effects we are seeing are attributable to linked mutations that happen to be enriched in high-density CpG regions of the genome, but most of the other classes of mutations (such as insertions and deletions) will decrease the amount of transcript (rather than increase it, as most methylation mutations are expected to do). We can exclude some of these mutations indirectly, because their mean effect on gene expression is known to be relatively large.[13] We divided the sample into those transcripts with a fold change greater than two and those less than two. The dataset with smaller effects contains 1,555 genes including 19,397 mutations, of which 1,450 are situated in high CpG sequence density regions.

Figure 5A shows that the D2:B6 ratio of transcript abundance is about 20-fold larger when mutations occur in regions of high CpG sequence density than in low density (mean ratio in high density: 0.19, mean ratio in low density: 0.01, t = 5.55, df = 1,662.6, p value = 3.3e−08). To examine whether CpG sequence density itself contributes to the increase in the D2:B6 ratio, Figure 5B plots D2:B6 ratios in high and low CpG sequence density for CpG sites *without* mutations.

There are higher D2:B6 ratios in the CpG high sequence density regions than in the low, and because there are so many more sites without mutations, the effect is highly significant (p = 4.11e−18), but the increase is much smaller than that found in regions with mutations (mean ratio in high: 0.034, mean ratio in low: −0.001). D2:B6 ratios in high CpG sequence density sites containing mutations are significantly higher than ratios in high CpG sequence density sites without mutations (t = 4.93, df = 1,493.1, p value = 9.18e−07), again consistent with the hypothesis that the effect of mutations in high CpG sequence density will be larger than in low CpG sequence density sites.

What is the effect of mutation on the transcript abundance ratio in the high- and low-density regions? We addressed this question using a linear model. In high-density regions, mutations have a highly significant positive effect (beta = 0.16, t = 5.2, p = 2.39e−07), while density makes no significant contribution (beta = −0.0002, t = −1.4, p = 0.16). The situation is reversed

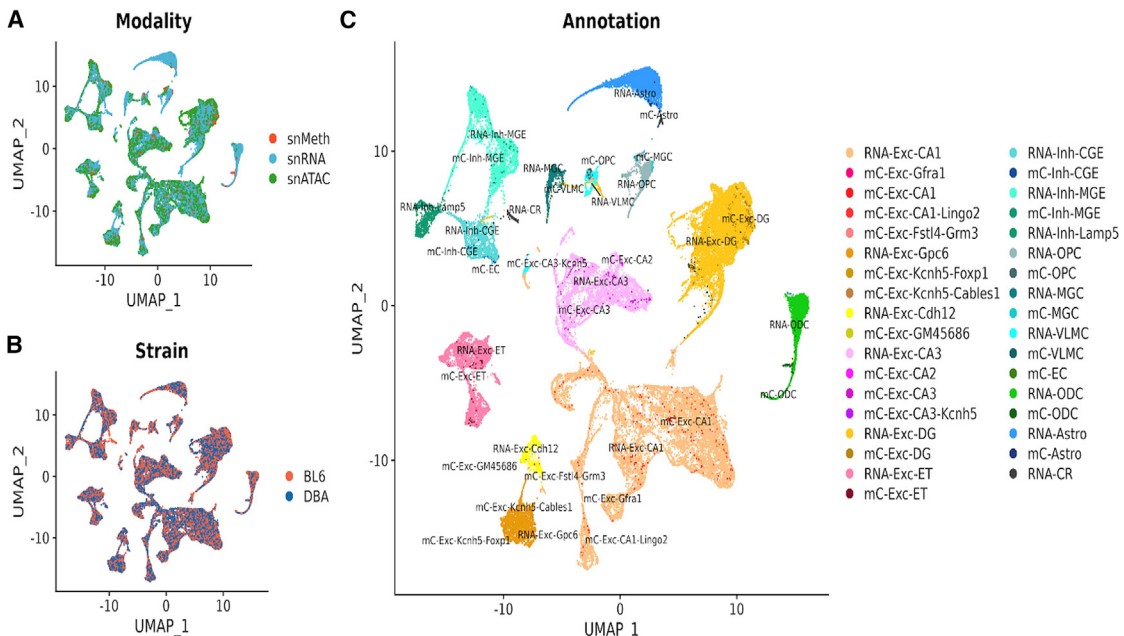

**Figure 4. Co-embedding of single-nucleus methylation with single-nucleus RNA and single-nucleus ATAC sequencing data for two strains, C57BL/6J and DBA/2J**

(A) UMAP embedding of single-nucleus methylation, single-nucleus RNA-seq, and single-nucleus ATAC cells after integration, colored by sequencing platform.

(B) is similar to (A) but colored by strain label.

(C) Concordance between independent single-nucleus methylation annotation and single-nucleus RNA transcriptomic-based annotation on major ventral hippocampus cell types.

in low CpG density regions: mutations make no detectable contribution (beta = 0.007, t = 0.911, p = 0.362), while the effect of CpG density, though small, is highly significant (beta = 0.006, t = 61.3, p < 2.2e−16). These results indicate that the effect of mutations altering methylation at CpG sites is undetectable when they occur in regions of low CpG density.

Might CpG sequence density be a surrogate for other confounds that explain the predictive power of mutation in the high-density regions? For example, it could be that high-density CpG sequences are enriched with functional elements that promote expression, and the mutations are linked to such elements. To exclude the effect of such confounds, we used a set of "universal chromatin state" annotations of the mouse genome based on over 900 datasets from various cell and tissue types.[45] We found a significant increase in the fit of a model that included an interaction between mutation and density (i.e., allowing the effect of mutation to vary according to CpG density), compared to a model that only included mutations, chromatin state, and density (high vs. low) to predict D2:B6 ratios (F = 28.7, p = 8.41e−08, and Table S4). Finally, we addressed the issue of whether p values were well calibrated for this model (and others) by randomly sampling the ratio and repeating the analyses 10,000 times (we did this by permuting expression levels among genes, STAR Methods).

What of the large effect changes? To test our expectation that the large fold changes in expression are likely *not* associated with changes in repressive methylation, we examined genomic regions associated with transcripts in which DEseq2 reports a change of more than 2-fold difference. We identified 216,168 po-

sitions associated with genes, including 4,693 mutations. Their effect is large but decisively negative: −0.23, p = 2.98e−23. This result justifies our exclusion of the larger effect loci from the analysis of the impact of mutations on methylated CpGs.

Our analyses make assumptions about the threshold for distinguishing high and low CpG sequence density regions and in the division of effects by fold changes in transcript abundance. We carried out analyses to test these assumptions and found that our main conclusions hold, namely the impact of mutations at methylated CpG sites depends on local CpG sequence density (STAR Methods, Figures S2, S3, S4, and S5 and Tables S3 and S4). We were also able to see the same effect in six other cell types for which we had sufficient data; for cell types with low coverage there were too few mutations lying within regions whose methylation state we could confidently call (Tables S5 and S6 and Figure S6).

### CpG mutations are enriched in enhancers

How might the CpG mutations in high-density regions be acting to alter RNA expression? Are there differences in the way mutations in low- and high-density regions operate? To address this, we examined the relationship between mutations and annotated functional elements. Since functional elements differ between cell types, we first generated a set of single-nucleus assay for transposase-accessible chromatin (ATAC) sequence data from the two strains (B6 and D2).

We generated a single-nucleus ATAC-seq dataset for a total of 27,206 nuclei (median read-pairs per cell: 29,806) from two biological replicates after doublet removal and QC (STAR Methods).

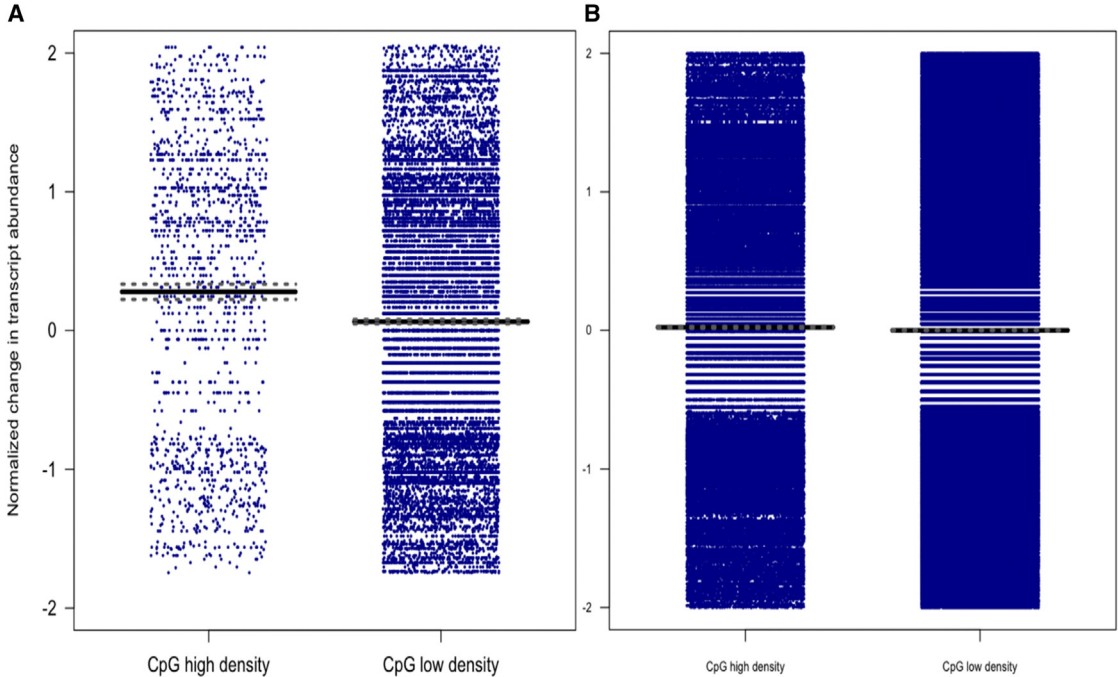

**Figure 5. Effect of mutations at methylated CpGs on D2:B6 ratios of transcript abundance in regions of low and high CpG sequence density**
Each dot represents the change in D2:B6 ratios of transcript abundance for mutations lying in regions of high CpG sequence density (>40 CpG/Kb) and in regions of low CpG sequence density (<40 CpG/Kb). The horizontal bars indicate the mean change in RNA transcript, with upper and lower 95% confidence intervals shown as dotted lines.
(A) Sites that have mutations disrupting methylated CpGs.
(B) Sites without mutations.

We then carried out integration and linking of our snATAC-seq dataset with our snRNA-seq dataset to obtain cell-type annotations Finally, we confirmed through joint embedding that major cell-type clusters within snATAC and snMethylation were highly consistent between modalities and strain (Figures 4A, 4B, and 4C and STAR Methods).

We annotated the open-chromatin sites based on their overlap with chromatin states.[45] We want to know if the location of CpG mutations that contribute to changes in transcript abundance differs in any of 15 chromatin state groups (excluding one state labeled "artifacts"), comparing sites lying in high CpG sequence density regions to those in low sequence density regions. For each site, we used a linear model, predicting the change in D2:B6 ratios from the presence of mutations, doing this separately in high and low CpG sequence density regions for each state. Table 1 shows the result, giving the number of sites for each chromatin state and the p value of the linear model for the predicted effect of CpG mutations in both high and low sequence CpG density regions of the genome. After correcting for multiple testing, only the effect of mutations in active enhancers in high CpG density regions is significant (p = 8.30E−04, exceeding a Bonferroni corrected threshold of 0.001 for testing 2 x 15 states).

## DISCUSSION

Our study of methylation states in eight inbred strains, combined with the nearly complete sets of sequence variants and func-

tional annotations, allows us to address the relationship between sequence and methylation variation. We find that the relationship depends in a non-linear fashion on the density of CpG sequences per kilobase of DNA. At densities less than 40 CpG/Kb, there are more mutations in the methylated than unmethylated DNA. The relationship is consistent across brain cell types. The significance of the inflection at 40 CpG/Kb is unclear, but presumably it reflects how cells determine CpG sequence density. It is known that increasing CpG sequence density alters promoter activity,[33] possibly by recruitment of ZF-CxxC domain-containing proteins that bind to unmethylated CpGs.[46]

At lower CpG sequence densities, cells compensate for the loss of methylated sites by recruiting additional sites in the same DNA segment, sites that arise due to mutational gain of cytosines. At higher sequence densities, the exact location of a methylated site becomes more important; specific sites are maintained even at the cost of reducing the number of methylated sites. In other words, the impact of a mutation on methylation depends on the CpG sequence density, with different consequences for embedded functional elements. These are exposed to higher mutation rates in lower density regions but with relaxed constraint on where the mutation will occur; in higher density regions, they are exposed to lower mutation rates, but mutations are likely to be constrained to preserve the same sites of methylation.

We tested this prediction by examining the impact of CpG density on mutations, using as output the relative change in

**Table 1. The association of CpG mutations on RNA transcript abundance varies by local chromatin state**

| Feature | High CpG density | | | | Low CpG density | | | |
|---|---|---|---|---|---|---|---|---|
| | No mut. | Mut. | Pct. | p value | No mut. | Mut. | Pct. | p value |
| Active enhancers | 17,109 | 200 | 1.155 | 8.30E−04 | 160,010 | 1,611 | 0.997 | 0.167 |
| Bivalent promoters | 11,970 | 39 | 0.325 | 0.238 | 19,974 | 64 | 0.319 | 0.842 |
| DNAase open chromatin | 11,241 | 164 | 1.438 | 0.201 | 102,697 | 1,223 | 1.177 | 0.608 |
| Heterochromatin | 921 | 16 | 1.708 | 0.278 | 20,419 | 330 | 1.59 | 0.379 |
| Polycomb repressed | 1,298 | 27 | 2.038 | 0.125 | 17,635 | 239 | 1.337 | 0.601 |
| Polycomb repressed and open chromatin | 3,996 | 42 | 1.04 | 0.027 | 11,060 | 111 | 0.994 | 0.385 |
| Promoter flank | 13,335 | 78 | 0.582 | 0.124 | 32,161 | 190 | 0.587 | 0.496 |
| Quiescent | 451 | 13 | 2.802 | 0.78 | 18,156 | 329 | 1.78 | 0.027 |
| Transcribed enhancers | 8,243 | 100 | 1.199 | 0.024 | 62,201 | 558 | 0.889 | 0.065 |
| Transcription | 8,342 | 131 | 1.546 | 0.986 | 105,918 | 1,116 | 1.043 | 0.072 |
| Transcription and exons | 10,546 | 61 | 0.575 | 0.417 | 41,331 | 236 | 0.568 | 0.143 |
| Transcription start sites | 10,274 | 16 | 0.155 | 0.662 | 26,832 | 53 | 0.197 | 0.192 |
| Weak enhancers | 5,139 | 87 | 1.665 | 0.573 | 93,371 | 1,172 | 1.24 | 0.102 |
| Weak transcription | 785 | 14 | 1.752 | 0.031 | 16,570 | 241 | 1.434 | 0.536 |
| Zinc finger genes | 633 | 9 | 1.402 | 0.496 | 4,456 | 53 | 1.175 | 0.767 |

The table shows 15 chromatin state groups (from Vu and Ernst[45]), together with the number of sites lying within genes (including 2 kb upstream and downstream) whose expression could be detected in the Exc-DG cell type in the hippocampus. The sites are divided into those lying in high and low CpG sequence density regions of the genome and then by whether they contain a mutation that disrupts a CpG site ("Mutation" and "No mutation" columns). The proportion (expressed as a percentage) of sites with mutations is shown for each chromatin state. The p value from a linear model testing for the predicted effect of mutations on RNA fold change is shown.

transcript abundance in a comparison of two inbred strains. Since CpG methylation in general suppresses transcription, we expected mutations that abrogate a methylated CpG site to increase the relative amount of transcript and found this to be so. Consistent with our predictions from how CpG density relates to methylation, we found that the impact of mutations depended on the CpG density. In regions of low CpG density, we were unable to detect a significant effect of mutation on transcription, despite the very large number of sites analyzed. By contrast, in regions of high CpG density, the far fewer mutations we found made a highly significant contribution to variation in transcript abundance (though the overall effect is small, less than 1%). We also found that in high-density regions, CpG mutations are enriched in chromatin states that mark enhancers.

It is commonly assumed that genetic effects on behavior (as with other complex traits) are likely mediated by changes in transcription, an idea supported by the success of transcription-wide association to facilitate the discovery of genes for common complex traits.[47] Assuming that the CpG mutations' effects on transcription can be extrapolated to their effects on behavior, our findings suggest ways to prioritize the detection of genetic variants that alter behavior. One implication is that variants within regions of low CpG sequence density are unlikely to have any detectable effect on a phenotype, which would simplify searches for causative mutations. Conversely, prioritizing the search for causative variants to regions of high CpG sequence density and to enhancers should accelerate the discovery of variants that are causal for behavior. Importantly, there are far fewer variants in high compared to low CpG sequence density regions of the genome (approximately 1,500 compared to 15,000 in a com-

parison of two strains). While still large, this number is easily within the range of massively parallel reporter assays.[48] Furthermore, when interest is focused on a single locus, identified for example from fine mapping of behavioral phenotypes, there will be far fewer candidate variants to consider: approximately three candidate variants will lie within a 5-Mb locus, mapped in a comparison between B6 and D2 strains (this does not take into account the non-random distribution of variants across the mouse genome).

Our results support those of others that the high mutation rates at methylated CpG sites depend in part on local sequence context and the genomic region.[43–54] We add to this literature by indicating that sequence variants disrupting methylation act primarily through a specific class of candidate functional elements (we identified enhancers) in the context of high CpG sequence density. While we are unable to distinguish standing genetic variation (subject to selective pressure) from *de novo* variation, which makes us unable to determine whether the pattern observed in mice is independent of selection, the pattern suggests that the increased mutational load associated with methylation may be targeted to a subset of functional elements, consistent with findings in other species.[55] In other words, it is possible that methylation may be being used to target increased rates of mutation to specific elements in the genome.

### Limitations of the study
There are several limitations of our study. First, we used two mouse replicates to generate the methylation RNA data, limiting our ability to detect differences between strains. Second, for most cell types, sequence coverage was too low to detect

most methylated sites. We cannot be certain that results found for the high-coverage cell types will apply to others, though as explained in the supplemental information, we think this is unlikely. Third, we cannot be certain we have identified all cell types. Cell types that we inadvertently believe to be homogeneous may confound our analysis, as the effects of mutations in cell-type-specific methylation sites will be obscured. We may also be missing cell types in which mutations have different effects from those documented here. Finally, our analysis is limited to CpG sites and to one region of the brain, the hippocampus. It is possible that mutations affecting non-CpG methylated sites behave differently. It is also possible, though we think unlikely, that different findings will emerge from analysis of different brain regions.

## STAR★METHODS

Detailed methods are provided in the online version of this paper and include the following:

- KEY RESOURCES TABLE
- RESOURCE AVAILABILITY
  - Lead contact
  - Materials availability
  - Data and code availability
- EXPERIMENTAL MODEL AND STUDY PARTICIPANT DETAILS
  - Mouse strains
- METHOD DETAILS
  - Ventral hippocampus microdissections
  - Generating snmC-seq2 libraries
  - Generating snRNA-seq libraries
  - Generating snATAC-seq libraries
- QUANTIFICATION AND STATISTICAL ANALYSIS
  - Mapping and primary quality control
  - Single-nucleus methylation data quality control and preprocessing
  - Single-nucleus RNA sequencing data quality control and preprocessing
  - Single-nucleus ATAC sequencing data quality control and preprocessing
  - Cell type cluster generation and modality integration
  - Analysis of cell type specific effects and relationships with sequence variation
  - Calibrating P-values
  - Testing for an interaction between CpG density and mutation

### SUPPLEMENTAL INFORMATION

### ACKNOWLEDGMENTS

This work was funded in part through NIH grants R01MH115979, R01MH125252, U01MH130995, and NIH DP1DA044371 and UCLA Jonsson Comprehensive Cancer Center and Eli and Edythe Broad Center of Regenerative Medicine and Stem Cell Research Ablon Scholars Program.

### AUTHOR CONTRIBUTIONS

C.L. performed the single-cell methylation analyses, data from which was processed by M.G.H. and Z.C. P.B.C. performed the snRNA and snATAC analyses with data processed by Z.C. J.F. designed the study and wrote the pipeline to identify the effect of mutations. J.M. provided statistical support. J.F. and E.M. co-wrote the manuscript, which was subsequently reviewed and edited by the rest of the authors.

### DECLARATION OF INTERESTS

The authors declare no competing interests.

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

## Article

CellPress

# STAR★METHODS

## KEY RESOURCES TABLE

| REAGENT or RESOURCE | SOURCE | IDENTIFIER |
|---|---|---|
| **Antibodies** | | |
| NeuN-488 | Millipore Sigma | MAB377X |
| NeuN-405 | Novus Biologicals | NBP1-92693AF405 |
| **Critical commercial assays** | | |
| Next GEM scATAC-Seq v1.1 | 10X Genomics | PN-1000175 |
| Chromium Next GEM Automated Single Cell 3′ Library and Gel Bead Kit v3.1 | 10X Genomics | PN-100014 |
| **Deposited data** | | |
| snmC-seq2 data of 8 mouse strains | This study | GSE245367 |
| snATAC-seq data of 2 mouse strains | This study | GSE245367 |
| snRNA-seq data of 2 mouse strains | This study | GSE245367 |
| Processed methylation data | This study | https://figshare.com/account/home#/collections/6943056 |
| Processed ATAC and RNA data | This study | https://figshare.com/account/home#/collections/6943059 |
| **Experimental models: Organisms/strains** | | |
| C57BL/6J | JAX | Strain ID: 000664 |
| DBA/2J | JAX | Strain ID: 000671 |
| CAST/EiJ | JAX | Strain ID: 000928 |
| FVB/NJ | JAX | Strain ID: 001800 |
| A/J | JAX | Strain ID: 000646 |
| WSB/EiJ | JAX | Strain ID: 001145 |
| PWK/PhJ | JAX | Strain ID: 003715 |
| BALB/cJ | JAX | Strain ID: 000651 |
| **Software and algorithms** | | |
| Cell Ranger V6.0.2 | 10X Genomics | https://www.10xgenomics.com/support/software/cell-ranger/downloads |
| Cell Ranger ATAC V2.0.0 | 10X Genomics | https://support.10xgenomics.com/single-cell-atac/software/downloads/latest |
| Bismark V0.20.0 | Krueger and Andrews[56] | https://github.com/FelixKrueger/Bismark |
| DESeq2 V1.34.0 | Love et al.[44] | https://bioconductor.org/packages/release/bioc/html/DESeq2.html |
| Seurat 4.0.5 | Stuart et al.[57] | https://github.com/satijalab/seurat |
| Signac 1.5.0 | Stuart et al.[58] | https://github.com/stuart-lab/signac |
| Model-based Analysis for ChIP-Seq (MACS) V3.0.0a7 | Zhang et al.[59] | https://github.com/macs3-project/MACS |
| DoubletFinder V2.0.3 | McGinnis et al.[60] | https://github.com/chris-mcginnis-ucsf/DoubletFinder |
| Scanpy V1.9.3 | Wolf et al.[61] | https://pypi.org/project/scanpy/ |
| Harmony V0.0.9 | Korsunsky et al.[62] | https://github.com/immunogenomics/harmony |
| SCTransform V0.3.2 | Hafemeister et al.[63] | https://github.com/satijalab/sctransform |
| All custom code used in this paper | This study | https://github.com/jonathanflint2/CodeForMethylationPaper |

## RESOURCE AVAILABILITY

### Lead contact

Further information and requests for reagents and resources should be directed to and will be fulfilled by the lead contact, Jonathan Flint (JFlint@mednet.ucla.edu).

 **CellPress**

**Cell Genomics**
Article

### Materials availability

This study did not generate new unique reagents.

### Data and code availability

- Raw and processed sequencing data generated for this study were deposited to NCBI GEO/SRA with accession number GSE245367 and are publicly available at the time of publication. Processed data relating to the results and method sections are shared in figshare
  - https://doi.org/10.6084/m9.figshare.23631984.v1
  - https://doi.org/10.6084/m9.figshare.23632044.v1
- All original code has been deposited at https://zenodo.org/records/10051912 https://github.com/jonathanflint2/CodeFor MethylationPaper and is publicly available as of the date of publication.
- Any additional information required to reanalyze the data reported in this paper is available from the lead contact upon request.

## EXPERIMENTAL MODEL AND STUDY PARTICIPANT DETAILS

### Mouse strains

All experimental procedures using live animals were approved by UCLA's Animal Care and Use Committee (protocol number ARC-2018-026). Male mice from eight inbred strains A/J, C57BL/6J, BALB/cJ, FVB/J, DBA/2J, WSB/EiJ, PWK/PhJ, and CAST/EiJ were purchased from Jackson Laboratories at 8 weeks of age and transferred to UCLA where they were kept for at least 7 days before tissue extraction. Animals were housed with *ad libitum* food and water in a 12 h light-dark cycle.

## METHOD DETAILS

### Ventral hippocampus microdissections

Adult male animals (Jackson Laboratories) were euthanized at 10–16 weeks old in an isoflurane chamber and decapitated. The brain was removed and the ventral region of the hippocampus was microdissected, snap frozen in dry ice, and stored at −80°C until processing. Tissue from ∼2 animals were combined into a single tube and considered a replicate, with 2 replicates per strain for snmC-seq2, snRNA-seq, and snATAC-seq experiments.

### Generating snmC-seq2 libraries

We carried out snmC-seq2 on microdissected tissue as previously described.[64] Briefly, frozen tissue was homogenized into single nuclei suspensions with Dounce homogenization, then immediately sorted on into a 384-well plate with a FACSAria sorter (BD Biosciences) at the UCLA Flow Cytometry Core. We selected for a 75-25 enrichment of neuronal vs. non-neuronal nuclei during FACS sorting using NeuN-488/DAPI counterstains (Millipore Sigma MAB377X). Bisulfite conversion and single-cell methylome libraries were generated following this step.

### Generating snRNA-seq libraries

Single nuclei suspension and library generation were completed at the Cedars Sinai Applied Genomics, Computation and Translational Core and followed the 10X protocol for the Chromium Next GEM Automated Single Cell 3′ Library and Gel Bead Kit v3.1 (cat# PN-100014) as described except for the following modifications:

Suspensions from cell nuclei were generated using the recommended method from the 10X scMultiome protocol (CG000375 Rev C) to lyse cells and obtain nuclei. Following single nuclei suspension generation, nuclei were counterstained for 7-AAD and NeuN-405 antibody (Novus Biologicals, 1:200) and sorted on a MACSQuant Tyto (Miltenyi Biotech) prior to GEM generation. We selected for a 75-25 split of NeuN+/7-AAD+ nuclei for neurons and NeuN-/7-AAD+ for non-neuronal nuclei respectively. We captured ∼10,000 nuclei per genotype per region per replicate on a single 10X GEM chip. All downstream library preparation was done according to the 10X Genomics protocol (CG000286) and sequenced on a Novaseq 6000 with a target of ∼40-50k reads per nucleus.

### Generating snATAC-seq libraries

Single nuclei suspension and library generation were completed at the Cedars Sinai AGCT core and followed the 10X protocol for Next GEM scATAC-Seq v1.1 (PN-1000175) as described except for the following modifications:

Nuclei suspensions were generated using the recommended method from the 10X scMultiome protocol (CG000375 Rev C) to lyse cells and obtain nuclei.

Following single nuclei suspension generation, nuclei were counterstained for 7-AAD and sorted on a MACSQuant Tyto prior to GEM generation. NeuN was not used for neuronal enrichment due to dye incompatibility between our NeuN antibody and a nuclear counterstain. After the sort, we carried out permeabilization of nuclei as per the protocol. We aimed to capture 10,000 nuclei per well x 8 wells, for a total of 80,000 nuclei over 8 total samples (∼10,000 nuclei per genotype per region per replicate). All downstream library preparation was done according to the 10X Genomics protocol (CG000209) and sequenced on a Novaseq 6000 with a target of >35k reads per nucleus.

## QUANTIFICATION AND STATISTICAL ANALYSIS

### Mapping and primary quality control

All reads were mapped to the mouse mm10, Genome Reference Consortium Mouse Build 38 (GCA_000001635.2)). The gene and transcript annotation used was a GENCODE GTF file.[65] snmC-seq2 reads were mapped using Bismark (V0.22.3) to SNP-swapped mm10 genomes. This was done to allow us to directly compare chromosomal coordinates between the 8 strains. SNPs for each strain were downloaded from the Sanger Institute Mouse Genomes website.[12] Custom code was used to generate 8 separate SNP-swapped genomes for each strain by replacing each corresponding SNP nucleotide position in the B6J mm10.fa file with the nucleotide of that strain.

snRNA and snATAC-seq reads were mapped using 10X Cell Ranger (V6.0.2) and 10X Cell Ranger ATAC (V2.0.0) respectively against mm10 for B6 and an SNP-swapped mm10 for DBA. We retained introns for RNA analysis while default settings were used for ATAC analysis.

### Single-nucleus methylation data quality control and preprocessing

Cells were filtered on the basis of several metadata metrics: (1) mCCC level <0.03; (2) global mCG level >0.5; (3) global mCH level <0.2; (4) total mapped reads >100,000; (5) Bismark mapping rate >0.5; and (6) percent genome covered >2.

Methylation features were calculated as fractions of methylcytosine over total cytosine across gene bodies ± 2kb flanking regions and 100kb bins spanning the entire genome. Methylation features were further split into CG and CH methylation types. Features overlapping our methylation mm10 blacklist were removed. 100kb bin features were then filtered on minimum mean coverage >500 and maximum mean coverage <3000. Gene body features were filtered on minimum coverage >5 and all remaining features were normalized per cell using the beta binomial normalization technique in allcools.[16]

### Single-nucleus RNA sequencing data quality control and preprocessing

All quality control and preprocessing were done under the Seurat package framework.[57] Per biological sample, we filtered out cells that (1) fall below the 5th percentile of the total UMI counts (nCount_RNA) or the 5th percentile total number of unique genes expressed (nFeature_RNA) or 700 unique genes expressed, whichever was more stringent; (2) are over the 95th percentile quantile in either the total UMI counts or the total number of unique genes expressed; (3) have larger than 5% mitochondria fraction (percent.mt).

Global coverage normalization: counts per million (CPM) was applied to each cell followed by log transformation ("LogNormalize"). We then projected cells from each biological sample to low dimensional space using principal components analysis (PCA) on highly variable features selected by Seurat. Potential doublets were identified and subsequently removed from the downstream analysis by DoubletFinder,[60] ran in the top 15 principal components space with the expected doublet rate set to the recommended amount from 10X genomics based on loading volume (10k nuclei per well).

### Single-nucleus ATAC sequencing data quality control and preprocessing

All quality control and preprocessing were done under the Seurat and Signac package framework.[58] Per sample, we first used Model-based Analysis for ChIP-Seq (MACS) to call sample specifc *de novo* peaks from its fragments file.[59] We then merged sample-specific sets of peaks to a unified peaks set while removing peaks with length larger than 10000bp or smaller than 20bp. A unified peaks by cells count matrix was constructed from the fragments file while removing cells with lower than 200 peaks detected and peaks only present in less than 10 cells. Cells were filtered based on the following criteria: (1) appropriate number of non-duplicate, usable read-pairs (passed_filters from Cell Ranger's output singlecell.csv). Specifically, we set it to larger than 3000, 4000, 2500, and 5000 for the 2 BL6 samples and 2 DBA samples respectively. (2) number of fragments overlapping peaks (peak_region_fragments from Cell Ranger's output singlecell.csv) falls within the 5th percentile and the 95th percentile. (3) ratio of fragments overlapping peaks over the total number of non-duplicate, usable read-pairs falls within the 5th percentile and the 95th percentile. (4) nucleosome_signal: the ratio of fragments between 147 bp and 294 bp (mononucleosome) to fragments <147 bp (nucleosome-free) is smaller than 4. (5) TSS enrichment score is larger than 2. We did not include a filter for ratio of peaks in black list regions over the total number of non-duplicate, usable read-pairs as this was removed during the construction of the DBA SNP-swapped reference genome.

We normalized the count data with Text Frequency Inverse Document Frequency (RunTFIDF) and performed Singular Value decomposition (RunSVD) on top 90% informative features selected by Signac (FindTopFeatures). The first low dimensional embedding was excluded from downstream doublet detection and clustering analysis due to high correlation with sequencing depth.

Potential doublets were identified and subsequently removed from downstream analysis by DoubletFinder,[60] ran on the $2^{nd} - 11^{th}$ low dimensional embedding with the expected doublet rate set to recommended amount from 10X genomics based on loading volume (10k nuclei per well).

Finally, we built a gene-by-cell transcriptional activity matrix that counts per cell, at the gene body and 2000bp upstream to capture the promoter region, the total number of ATAC-seq counts.

## Cell type cluster generation and modality integration
### Single-cell methylation clustering

Principal component analysis was run using Scanpy[61] default parameters followed by k-nearest neighbors (knn) using only the top 16 principal components by amount of variance explained and k = 15. Divergences between strains were evident in initial unsupervised clustering so Harmony[62] was used to correct the batch between strains and unbias clusters arising from strain differences. After Harmony was applied, iterative clustering was performed with a combination of leiden unsupervised clustering and UMAP dimensionality reduction, identifying clusters as cell types by marker gene hypomethylation.[16]

### Single-nucleus RNA sequencing data integration, clustering and annotation

Gene counts were normalized using SCTransform,[63] and regressed out percentage of reads from mitochondrial genes. We then integrated cell from all samples using reciprocal principal components analysis (rPCA) implemented in Seurat 4.0.5[57] on the top 5000 integration genes and top 30 reciprocal principal components. For clustering, we standardized the integrated data, performed PCA on all integrated genes and ran *de novo* Louvain clustering algorithm in the top 15 principal components space with resolution set to 0.1. Cluster markers that are conserved between the strains were called using non-parametric Wilcoxon rank-sum test and subsequently used for annotation. We annotated clusters by manually checking conserved markers against the ALLEN BRAIN MAP's Mouse Whole Cortex and Hippocampus dataset.

### Single-nucleus ATAC sequencing data integration, clustering and annotation

We first jointly projected all cells' ATAC peak profile to uncorrected Latent Semantic Indexing (LSI) embeddings with TFIDF transformation followed by calculating SVD on the top 90% most informative peaks. Peak profile embeddings were then integrated in shared low dimensional space via integration anchors identified in the 2nd to 30th reciprocal LSI space as implemented in Signac 1.5.0. We then integrated cell transcriptional activity profiles by performing SCTransform after regressing out percentage of activity from mitochondrial genes and carried out rPCA integration on integration genes identified from the single-nucleus RNA experiment and the top 10 reciprocal principal components. We transferred the single-nucleus RNA annotation onto the single-nucleus ATAC cells by linking the RNAs expression profiles with ATAC transcriptional activity profiles through canonical correlation analysis (CCA) described in Seurat. Pairs of cells from each modality that are mutual nearest neighbors in the top 15 canonical component space were identified as "transfer anchors". "Anchors" were further filtered and weighted by distances in the integrated peak embeddings prior to impute ATAC cells' annotation.

### Co-embedding single-nucleus methylation with single-nucleus RNA and single-nucleus ATAC sequencing data

While we obtained separate independent annotations for single-nucleus methylation cells and RNA and ATAC cells, a joint embedding demonstrates that cluster annotation is highly concordant across modalities.

We used the negative of the average mCH fraction of the gene body ± 2kb as the proxy of methylation cell's transcriptional activity as described previously.[16] Single-nucleus RNA expression profiles were linked to single-nucleus methylation gene body mCH profiles via CCA on RNA integration genes. We then identified "transfer anchors" in the top 15 canonical component space and used them to impute methylation cells' expression profiles. Single-nucleus ATAC cell expression profiles were imputed similarly with previously computed "anchors". All three modalities were merged on their integrated or imputed expression profiles and projected to low dimensional space via PCA, and visualized by UMAP performed on the top 15 principal components.

In general, cell-type annotation demonstrated high concordance across the three modalities, except on single nucleus methylation of non-neuronal cells, where the VLMC, OPC and MGC cell clusters did not colocalize with the corresponding expression-based annotation counterpart. This could be partially explained by the fact that mCH methylation is largely not present in the non-neuronal population.[66] In addition, both the "transfer anchor" and subsequently the imputed transcriptomic profile for single nucleus methylation cells were identified using the "gene body only" mCH fraction, a significant reduction in information compared to what was used for the *de novo* single nucleus methylation annotation, which included both the mCG and mCH fraction at 100kb genome wide.

## Analysis of cell type specific effects and relationships with sequence variation
### Cell-type specific differential test for single-nucleus ATAC and single-nucleus RNA

We used DESeq2[44] for cell-type specific pseudobulk level differential expression analysis and differential accessibility analysis. Per cell type, raw counts were aggregated to replicate level and DESeq2 was run under default parameters to detect statistical evidence of strain differences.

### Pipeline to identify relationship between methylation and sequence variation in multiple strains

*Data from each strain for each cell type was combined into a single R data frame, where the following pieces of information were included for each site.*

| chr | Chromosome | |
|-----|-----------|-----|
| pos | bp position | |
| aj.mc | Methylated reads | A/J |
| aj.cov | Total reads | A/J |

*(Continued on next page)*

*Continued*

| chr | Chromosome | |
|-----|------------|---|
| b6.mc | Methylated reads | C57BL/6J |
| b6.cov | Total reads | C57BL/6J |
| balb.mc | Methylated reads | BALB/cJ |
| balb.cov | Total reads | BALB/cJ |
| cast.mc | Methylated reads | CAST/EiJ |
| cast.cov | Total reads | CAST/EiJ |
| d2.mc | Methylated reads | DBA/2J |
| d2.cov | Total reads | DBA/2J |
| fvb.mc | Methylated reads | FVB/J |
| fvb.cov | Total reads | FVB/J |
| pwk.mc | Methylated reads | PWK/PhJ |
| pwk.cov | Total reads | PWK/PhJ |
| wsb.mc | Methylated reads | WSB/EiJ |
| wsb.cov | Total reads | WSB/EiJ |
| strand | Strand | |
| type | CG or CH | |
| total.mc | Total methylated reads for all strains | |
| total.cov | Total coverage over all strains | |
| fraction.mc | Fraction of methylated reads | |
| n.strains | Number of strains for which data were available | |
| var.pos | Postion of variants with respect to the position of the methylated site | |
| var.sdp | Strain distribution pattern of sequence variation | |

This information was incorporated using custom perl scripts run in the following order, for each chromosome

(1) FindMethylatedCoordinatesForAllStrainsInOneTissue.pl -f <file of filenames of pseudobulk files (divided by chromosome) >
(2) GetAllMethylatedSitesFromCoordinates.pl -C chromosome -c <coordinate file> -f < output of step 1> > chr.n.cell.type.file
(3) IdentifyVariableMethylation.pl -f chr.n.cell.type.file > chr.n.cell.type.annotated.txt
(4) MatchToGenes.pl -c chr.n.cell.type.annotated.txt -g refGene.txt -i > chr.n.cell.type.annotated.genes.txt
(5) AddVariantsToDiffFile8Strains.pl -v chr.n.seq.vars -f chr.n.cell.type.annotated.genes.txt -a -c -w 0 > chr.n.cell.type.annotated.genes.variants.txt

Genome location was obtained from mm10, Genome Reference Consortium Mouse Build 38 (GCA_000001635.2)) and gene information from the associated databases.[65] Sequence variants were downloaded from the Sanger Institute Mouse Genomes project.[12]

We confirmed that the sequence variants coincided with the expected position by searching for sites with rs numbers and checking that the coordinates agreed between those in the file and those in the mouse mm10 assembly. We confirmed that a file has the correct sequence variants by downloading genome sequence from the UCSC browser and confirming that the sequence is "C" at the "+" strand indices from the comparison file. We added to this file the location of all CpGs, from the genome and from each strain, with the script:

FindCpGs.pl -c chromosome (assumes the presence of a sequence file to process of the form chr1.fa in a given directory) -f <annotation file>

This script identifies a five base pair context around any cytosine in the genomic sequence, testing for the presence of any variant that changes the reference B6 genome to a C by using the strain variant information.[12] This generates a file with the 5 bp sequence context in the reference genome, the sequence for each strain (marked as CG for CpG), the number of methylated sites, the sequence coverage and a column for annotations (containing gene information and other potentially relevant features, downloaded from UCSC genome browser databases.[65]

We extracted regions of 2 kilobases in length (different segment sizes were also analyzed), examined for each the methylation and mutational spectrum. To do so we consider variation in coverage in each segment and correct for this by down sampling the strain with the most sequence reads at each region (we randomly reduced the amount of methylation proportionate to the ratio in the total sample). We require a minimum coverage of 5 reads and to call a site methylated we require at least 10% of the reads to be methylated. If the coverage is good enough to call a site, but lacks evidence for methylation, then the site is regarded as not methylated.

We compared two definitions of similarity

(1) - two regions are the same if they contain the same amount of methylation

(2) - two regions are the same if they have the same pattern of methylation (i.e., the same sequence cytosine residues are methylated)

In definition 1 there can be different sites that are methylated but the total is the same - so definition 2 is a subset of definition 1.

We ask whether the amount of methylation in a region is maintained to overcome the impact of mutation (if this is true, there will be more mutations separating CAST from B6 than from BALB and B6 but the methylation will be the same (by definition 1 above)). To get information about the pattern and the equivalence of methylation we derive two measures for each segment for each pair of strains: the probability that any pair of sites is in the same state (methylated or not) and the ratio of the total number of methylated sites in the two strains.

We run a pairwise strain comparison in which we count the following: the number of sites that are CpG, the number of sites that are methylated, the number of sites that are unmethylated, the number of methylated CpG sites that are mutated and the number of unmethylated CpG sites that are mutated.

We estimate the probability that two sites are equal (either 0, 0 or 1,1) using the following algorithm in perl:

```perl
sub prop_occurrence {
my ($array1_ref, $array2_ref) = @_; my $p = 1; my $q = 0;
# Convert binary data to numeric data
my @array1 = map { $_ ? 1 : 0 } @$array1_ref; my @array2 = map { $_ ? 1 : 0 } @$array2_ref;
# Count the occurrences of p and q in array1
my $count_p_array1 = grep { $_ == $p } @array1; my $count_q_array1 = grep { $_ == $q } @array1;
# Count the occurrences of p and q in array2
my $count_p_array2 = grep { $_ == $p } @array2; my $count_q_array2 = grep { $_ == $q } @array2;
# Count the occurrences of p and q at the same site in both arrays
my $count_pp = 0;
my $count_qq = 0; for my $i (0..$#array1) {
$count_pp++ if ($array1[$i] == $p && $array2[$i] == $p);
$count_qq++ if ($array1[$i] == $q && $array2[$i] == $q);
}
# Calculate the proportions
my $prop_p_array1 = $count_p_array1/scalar(@array1);
my $prop_p_array2 = $count_p_array2/scalar(@array2);
my $prop_q_array1 = $count_q_array1/scalar(@array1);
my $prop_q_array2 = $count_q_array2/scalar(@array2);
my $prop_pp = $count_pp/scalar(@array1);
my $prop_qq = $count_qq/scalar(@array1);
return ($prop_p_array1, $prop_p_array2,$prop_q_array1,$prop_q_array2, $prop_pp, $prop_qq);
}
```

These analyses are performed using the perl script

```
PairwiseComparison.pl -f <file>
```

The output contains the CpG density for the segment analyzed, probability and ratios, together with the numbers of each category used to generate those results.

Finally, we sum results across segments with the same CpG density with the script.

```
SumByDensity.pl -f <file>
```

This script works out the probability and ratio for all sites in each segment, using the same approach as described above. The script outputs the following information: 'seg', the CpG density, 's1' and 's2' the two strains compared, 'CG.all' the number of CpG sites, 'CG.me.0' the number not methylated,'CG.me.1' the number methylated, 'mutant.me.0' the number of mutant sites not methylated, 'mutant.me.1' the number of methylated mutant sites, 'prob' the probability that sites are in the same methylation state, 'ratio' the ratio of the number of methylated sites, and the number of methylated sites in each strain('s1.me' and 's2.me'). These files are used to generate the figures in the main text and supplemental material. The script for these plots is MethylationSequenceComparison.R.

***Assessing whether mutations have an effect dependent on CpG density***

We combined information from the different modalities into one text file with this script

```
CombineModalities.pl -d density file -r rna DESEQ2 file -R RNA count file -a atac DESEQ file2 -A atac count file
-f counted file -c chromosome
```

This includes heterochromatin states information[45] which is included in a column in the output file. Subsequent analyses are performed in R.

Predicted phenotypes (RNA fold change) are quantile normalized with this function

```
invnorm <- function (x) {
y = (rank(x,na.last = "keep")-0.5)/sum(!is.na(x))
```

```
return (qnorm(y))
}
```

We compared differences between mean effects of fold change in the two groups (high and low CpG density using a t test

```
t.test (data$normalized.log2fold[data$high = = 1],data$normalized.log2fold[data$high = = 0])
```

We ran a series of linear models to test the effect of mutations on fold change in the transcript abundance. These models are of the form:

```
summary (lm(invnorm(rna.log2fold) ~ mutation * density, data = data))$coefficients.
```

For assessing the impact of annotations we establish a null model in which annotations, density (high vs. low) and mutations predict change in transcript abundance

```
fit0 = lm (normalized.log2fold ~ chromosome + position + annotations + density + mutations, data = data)
```

We compare this null model with one that allows an interaction between mutation and density

```
fit1 = lm (normalized.log2fold ~ chromosome + position + annotations + density + mutations + density:mutations, data = data)
```

and compare the fit of the two in an analysis of variance

```
anova (fit0, fit1)
```

### Calibrating P-values

We explored the distribution of P-values using the 'sample' function in R. We re-sampled at the gene level for the expression data (so each gene is re-assigned an expression level from a different gene) and examined the P-value distributions to determine if the P-values were well calibrated.

### Testing for an interaction between CpG density and mutation

Our hypothesis is that the effect of mutation on transcript abundance depends on CpG sequence density in a non-linear way (larger in regions of high density and lower in regions of low density). In the main text we test this by looking at the impact of features separately in high and low sequence density regions. To examine the involvement of covariates we use a slightly different approach: we test for the presence of an interaction between mutations and 'high' versus 'low' density region. While the interpretation of the effect size of the interaction is not intuitive, the test has the virtue of delivering a single result for answering our question while including the impact of covariates that could confound the interpretation.

We tested for an interaction between mutation and density, including the effect of chromatin states, using a set of mouse 'universal' chromatin states.[45] We assigned chromatin states to specific cell types by searching for an overlap between ATAC sites and annotations, requiring a total of at least 10 reads in the B6 and D2 strains. The model we tested, in R formulation, is:

```
lm (normalized.log2fold ~ annotations + density + mutations + density:mutations, data = data)
```

Results, shown in Table S2, reveal a highly significant interaction: $p = 8.41E{-}08$.

We used the same model to test the sensitivity of our results to assumptions about CpG sequence density and expression changes. We explored the impact of the thresholds we used for defining high and low CpG sequence density and the impact of excluding, or including, sites on the genome based on gene expression levels.

To examine the threshold for distinguishing high and low sequence density CpG regions of the genome, we divided CpG sequence density into high and low regions based on the results of Figure 2 in the main text, choosing the inflection point of the curve at 40 CpG/Kb. What happens if a different threshold is chosen? We include all transcripts, regardless of the size of the DESeq estimated log2fold change. We choose CpG sequence density thresholds between 14 and 60, and for each one divided the sample into high and low sequence density regions. We then calculated the interaction between the mutation and the RNA abundance, using the same linear model described above. Figure S2 shows the results, demonstrating a peak in the effect size at a density of 38 CpG/Kb.

What happens if we restrict analysis to RNA transcripts where there is some evidence of a difference between the two strains and where the effect is small (less than a 2-fold change)? Figure S3 gives the answer: the peak effect size is at 38, which coincides with the most significant interaction. The main difference is that the effect estimates are much bigger, as noted in the main text.

What is the effect of altering our threshold for including transcripts in the analysis? In the main text we describe results where we exclude transcripts with log2 fold values greater than 2. We have no prior evidence to support choosing this value, so what happens if we alter it? How robust are the results? We chose a CpG sequence density threshold of 38 and ran the interaction analysis thresholding transcripts on log2 fold values between 0.2 and 8. The effect size of the interaction is shown in Figure S4, and data in Table S3. The effect becomes positive at values greater than 0.5, and has a maximum at a threshold of 2, drops to just over 0.1 by a threshold of 4, and remains stable thereafter. Since the accuracy of effect size estimates will vary considerably depending on read counts and the true difference between the strains, we also stratified by the P-value of the DESeq analysis. Figure S5 shows the same pattern for the change in interaction effect size (a logP of 10 is approximately the same as a two -old change in the ratio). We also tested the effect of altering P-values thresholds for including transcripts in our analysis (in the main text we report results for including transcripts where the P-value for the differential expression, obtained from DESeq2, was less than 0.05). For each threshold we ran an interaction analysis, testing the dependence on the high density of CpG sites of the relationship between normalized change in transcript abundance and the presence of mutations. The results, shown in Table S4, demonstrate that the interaction effect is robust to the threshold we use.

