## [Data S1. Transparent peer review records for Flint et al. · Cell Genomics]

Title and authors

Summary

Initial submission: Received : 7/10/2023

Scientific editor: Laura Zahn

First round of review: Number of reviewers: 2
Revision invited : 8/17/2023
Revision received : 9/8/2023

Second round of review: Number of reviewers: 2
Accepted : 11/6/2023

Data freely available: Yes

Code freely available: Yes

This transparent peer review record is not systematically proofread, type-set, or edited. Special characters, formatting, and equations may fail to render properly. Standard procedural text within the editor's letters has been deleted for the sake of brevity, but all official correspondence specific to the manuscript has been preserved.

Referees' reports, first round of review

Reviewer #1: Comments enter in this field will be shared with the author; your identity will remain anonymous.

In this study, the authors generated single-cell multi-omics sequencing data in eight mouse strains, including the DNA methylation, chromatin accessibility and transcriptome of the hippocampus tissue. They found a non-linear relationship between DNA sequence and methylation variations, which was associated with the density of CpGs and could be used to prioritize the vital variants. Besides, they found CpG mutations were frequently enriched in the active enhancers. The number of input single cells were large, however, the number of cells for DNA methylation in each replicate was limited, which may introduce potential bias and dampen the integrated analysis combined with the chromatin accessibility and transcriptome. Additionally, while they captured cells from all eight strains and identified cell types, the downstream analysis was merely focused on limited cell types and performed cross-strain comparison mainly between two mouse lines. Nevertheless, this manuscript is falling short for experimental validation, mechanistic insights, and causality with phenotype outcomes. As the authors mentioned in the Discussion section, it's not a novel finding that high mutation rate in low to intermediately methylated CpG sites. Similarly, the enrichment of CpG mutations in somatic enhancers based on ATAC-seq data was not surprising, as this technique is used to identify cis-regulatory elements like promoters and enhancers. It should also be examined the mutations in the genome-wide context. For instance, are the mutations favorably enriched in repetitive elements? Overall, the results are deemed to be preliminary, typos are always appeared in the manuscript, and the conceptual advance is limited.

Reviewer #2: Flint et al. performed single cell methylation analysis of hippocampus tissue of 8 mouse strains, and assess the correlation between DNA methylation density and mutations in the genome. It is an interesting study, which highlight the view that the distribution of mutations is not random in the genome. However, there are several limitations that needed to be addressed.

- 1) Genome coverage of snmC-seq in each single cell should be provided. I wonder whether the coverage can affect the methylation level within the indicated density of CpGs.
- 2) Which tool is used for mapping snmC-seq data? In the method, different descriptions are provided, "snmC-seq2 reads were mapped using STAR aligner to SNP-swapped mm10 genomes." "snmC-seq2 reads were mapped using Bismark (V0.22.3)."
- 3) Are there methylation differences between the classical laboratory strains and wild-derived inbred strains?
- 4) Only $P < 0.05$, no adjusted P or FDR, was used to identify transcripts that were differentially expressed in the DESeq2 analysis?
- 5) Please acknowledge the limitation of your study in the Discussion section.
- 6) Please check your language carefully, for example "and stored at -80 until processing."

Reviewer #3: The study conducted by Flint et al. investigated the influence of DNA methylation on sequence variants through the generation of comprehensive, genome-wide, base-pair resolution maps across various cell types from eight inbred mouse strains. Initially, the authors performed single nucleus methylation sequencing on hippocampal samples and examined the correlation between methylation and CpG density. Moreover, they explored the impact of CpG mutations on gene expression by integrating single-cell RNA data. Additionally, the researchers employed single nucleus ATAC sequencing to elucidate the connection between CpG mutations and functional genomic elements. The findings suggest the impact of sequence variation depends on local CpG sequence density as mediated by DNA methylation, particularly in enhancer regions. The authors also shared their bioinformatics code and presented their work coherently. However, there are several noteworthy aspects that warrant attention:

1. Supplementary materials demonstrate the effect of CpG density on multiple cell types, including Exc-CA. Given the size of Exc-Cas is similar to Exc-ENT (Figure 1), it would be beneficial for the authors to further explore why Exc-CA is different. Would it be possible to investigate the influence of density by combining all the Exc-CA subclusters?
2. Despite the utilization of eight inbred mouse strains in this investigation, the sample size is confined to duplicates. To provide a comprehensive view, the authors should acknowledge this limitation in the discussion section.
3. The method of fluorescence-activated nuclear sorting was employed to isolate NeuN-positive and NeuN-

negative populations for snmC-seq. To enhance clarity, the authors should specify whether this sorting step was implemented in sn-RNAseq or sn-ATACseq

4. The authors used a liberal threshold of $p < 0.05$ (unadjusted) to identified 2049 transcripts that are differentially expressed. How many of these passed $p < 0.05$ (adjusted)? How would the results change when adjusted pvalue is used.

5. In the supplementary, the authors looked into the effect of density in multiple cell types but didn't address the association of CpG mutations on RNA transcript abundance depending local chromatin state.

Authors' response to the first round of review

Reviewer #1:

In this study, the authors generated single-cell multi-omics sequencing data in eight mouse strains, including the DNA methylation, chromatin accessibility and transcriptome of the hippocampus tissue. They found a non-linear relationship between DNA sequence and methylation variations, which was associated with the density of CpGs and could be used to prioritize the vital variants. Besides, they found CpG mutations were frequently enriched in the active enhancers. The number of input single cells were large, however, the number of cells for DNA methylation in each replicate was limited, which may introduce potential bias and dampen the integrated analysis combined with the chromatin accessibility and transcriptome.

While we accept that we used only two biological replicates, we do not believe that this has introduced bias. Critically, the sequence coverage (reflecting the number of cells) we obtained for 12 cell types was greater than 10X and for the Exc-DG cell type more than 50X. For all these cell types we observe the same relationship between mutations, CpG density and methylation.

Reviewer 2 also raised concern about coverage. In our revised manuscript we show that coverage does not influence the relationship between methylation and CpG density. We have included an additional supplemental figure (now S2) and have added a sentence to the text of page 7 of the manuscript "The same relationship is observed for other cell types for which we have sequence coverage greater than 10 fold. At lower sequence coverages the estimates of percentage methylation for most of the CpG density segments is too variable to reveal the relationship (illustrated in Supp Fig 2)." We include a full set of figures for all cell types (i.e. with varying degrees of coverage), which is included below in our response to reviewer 2.

We have also included a section in the discussion that presents limitations to our study, including the use of two replicates

Additionally, while they captured cells from all eight strains and identified cell types, the downstream analysis was merely focused on limited cell types and performed cross-strain comparison mainly between two mouse lines.

Downstream analysis of the methylation data was performed for all eight strains. We included RNA and ATACseq data to test specific hypotheses derived from findings in the eight way strain analyses. We don't think this is a limitation of our findings

Nerveless, this manuscript is falling short for experimental validation, mechanistic insights, and causality with phenotype outcomes. As the authors mentioned in the Discussion section, it's not a novel finding that high mutation rate in low to intermediately methylated CpG sites. Similarly, the enrichment of CpG mutations in somatic enhancers based on ATAC-seq data was not surprising, as this technique is used to identify cisregulatory elements like promoters and enhancers. It should also be examined the mutations in the genomewide context. For instance, are the mutations favorably enriched in repetitive elements?

We agree with the reviewer about the lack of novelty in these two areas, but this was not a take-home message of our paper. Our observation is that the effect of mutations altering methylated CpG sites depends on CpG density (not CpG methylation). We examined this extensively, across the genome. The CpG mutations were not enriched in any specific class of repetitive element.

Overall, the results are deemed to be preliminary, typos are always appeared in the manuscript, and the conceptual advance is limited.

We have corrected typographical errors. We believe that we have made an important conceptual advance in

identifying the value of prioritizing a class of mutations (those that affect CpG sites lying within regions of high CpG density).

Reviewer #2:

Flint et al. performed single cell methylation analysis of hippocampus tissue of 8 mouse strains, and assess the correlation between DNA methylation density and mutations in the genome. It is an interesting study, which highlight the view that the distribution of mutations is not random in the genome. However, there are several limitations that needed to be addressed.

1) Genome coverage of snmC-seq in each single cell should be provided. I wonder whether the coverage can affect the methylation level within the indicated density of CpGs.

We have now included data on genome coverage in a new table in Supplemental material. The high coverage in Exc-DG cell type allows us to capture almost all methylated sites and we used these results as the basis for our conclusions. However, the same relationships are observed at lower coverage.

To demonstrate this, we plotted the relationship between density of CpGs and methylation level for each cell type, each of which has a different coverage. For cell types with sequence coverage less than ten, the relationship between CpG density and methylation is unclear because of the large variation in the estimates of methylation for most units of CpG density. As coverage increases the pattern becomes clearer. We have included an additional supplemental figure (now S2) to show three examples. We have added a sentence to the text of page 7 of the manuscript "The same relationship is observed for other cell types for which we have sequence coverage greater than 10 fold. At lower sequence coverages the estimates of percentage methylation for most of the CpG density segments is too variable to reveal the relationship (illustrated in Supp Fig 2)."

We add here a complete set of figures for the reviewer to consider. These could be included in supplemental material, although we feel the point is adequately made with the figure showing three cell types. In the figure below, cell types are arranged in order of sequence coverage from 7.2 up to 51.

2) Which tool is used for mapping snmC-seq data? In the method, different descriptions are provided, "snmCseq2 reads were mapped using STAR aligner to SNP-swapped mm10 genomes." "snmC-seq2 reads were mapped using Bismark (V0.22.3)."

We apologize for this error in the method section. snmC-seq2 reads were mapped to SNP swapped mm10 reference genome using Bismark (V0.22.3). We have made this correction in the Methods section of the paper.

3) Are there methylation differences between the classical laboratory strains and wild-derived inbred strains?

There are differences between pairs of strains, including between classical laboratory strains and wild-derived inbred strains. However, overall methylation states correlate highly, as shown in the pairwise correlation coefficients for the Exc-DG hippocampal cell type below. Correlations in methylation states between classical

laboratory strains and the WSB wild-derived inbred strain are similar to those among the classical laboratory strains (0.91); the correlations between the PWK wild-derived inbred strain and classical laboratory strains is similar to that with Castaneus (0.83)

	A	B6	BALB	CAST	DBA/2	FVB	PWK
B6	0.915						
BALB	0.936	0.917					
CAST	0.834	0.829	0.828				
DBA/2	0.923	0.906	0.912	0.836			
FVB	0.934	0.905	0.912	0.825	0.911		
PWK	0.842	0.826	0.838	0.848	0.839	0.821	
WSB	0.924	0.903	0.91	0.842	0.909	0.905	0.842

4) Only $P < 0.05$, no adjusted P or FDR, was used to identify transcripts that were differentially expressed in the DESeq2 analysis?

We now include results for a series of P-values thresholds to demonstrate the robustness of our results to the threshold applied. For each threshold we ran an interaction analysis, testing the dependence on the high density of CpG sites of the relationship between normalized change in transcript abundance and the presence of mutations.

The table below summarizes the findings and is now included in the Supplemental Material in the section entitled "Sensitivity to assumptions about CpG sequence density and expression changes"

LogP threshold	Interaction effect	Interaction pval	Interaction logp
0.5	0.05	1.49E-05	4.83
1	0.06	1.87E-09	8.73
2	0.07	3.17E-12	11.5
3	0.07	8.14E-15	14.09
4	0.07	1.66E-13	12.78
5	0.07	5.98E-15	14.22
6	0.07	1.47E-13	12.83
7	0.07	4.51E-13	12.35
8	0.06	4.78E-12	11.32
9	0.07	4.78E-15	14.32
10	0.08	9.25E-17	16.03
11	0.08	7.70E-17	16.11
12	0.08	4.47E-17	16.35

5) Please acknowledge the limitation of your study in the Discussion section.

We added a section to the discussion which discusses the limitations of our study, including the use of duplicates, and the low coverage for some cell types

6) Please check your language carefully, for example "and stored at -80 until processing."

We have checked the manuscript for typographical and other errors. We have corrected the phrase noted above to "and stored at -80°C until processing."

Reviewer #3:

The study conducted by Flint et al. investigated the influence of DNA methylation on sequence variants through the generation of comprehensive, genome-wide, base-pair resolution maps across various cell types from eight inbred mouse strains. Initially, the authors performed single nucleus methylation sequencing on hippocampal samples and examined the correlation between methylation and CpG density. Moreover, they explored the impact of CpG mutations on gene expression by integrating single-cell RNA data. Additionally, the researchers employed single nucleus ATAC sequencing to elucidate the connection between CpG mutations and functional genomic elements. The findings suggest the impact of sequence variation depends on local CpG sequence density as mediated by DNA methylation, particularly in enhancer regions. The authors also shared their bioinformatics code and presented their work coherently. However, there are several noteworthy aspects that warrant attention:

1. Supplementary materials demonstrate the effect of CpG density on multiple cell types, including Exc-CA. Given the size of Exc-Cas is similar to Exc-ENT (Figure 1), it would be beneficial for the authors to further, m2q =564' bining all the Exc-CA subclusters?

We agree that it is worth exploring why Exc-CA1 is different. We showed in Supplemental material the effect in Exc-CA1 was negative (opposite in effect to other cell types), and we supposed that this might be because the cell type wasn't pure. We supported this hypothesis by looking to see what happened when all cell types were combined. As predicted, in the joint cell type, the interaction effect is opposite to that predicted for mutations that relieve suppression. However, this observation does not prove that Exc-CA1 is a composite cell type.

We revisited this question by using the RNAseq data from strains C57BL/6J and DBA2/J to subdivide Exc-CA1. By jointly embedding the modalities we identified three subtypes within the larger Exc-CA1 cell types, which are now included in our list of recognized cell types. By subdividing the sample size drops and we lose power, but the effect of the mutations is still observable for the two cell types with the largest coverage (Fras1 and Galnt16), if not significant, as shown in the figure below. We have included this figure and the results of the t-tests in Supplemental material, and have removed the now redundant analysis of bulk tissue

2. Despite the utilization of eight inbred mouse strains in this investigation, the sample size is confined to duplicates. To provide a comprehensive view, the authors should acknowledge this limitation in the discussion section.

We include in the discussion a section describing the limitations of the study, noting that we used only duplicates

3. The method of fluorescence-activated nuclear sorting was employed to isolate NeuN-positive and NeuN-negative populations for snmC-seq. To enhance clarity, the authors should specify whether this sorting step was implemented in sn-RNAseq or sn-ATACseq

We indicate in the Methods that both snRNA and snATAC were done with a FANS step prior to droplet/GEM generation. In our scRNA experiment, we selected for NeuN+ neurons in a similar ratio of neurons/non-neurons as the methylation experiment (75-25). However, for the snATAC experiment, NeuN was not used for neuronal enrichment due to dye incompatibility between our NeuN antibody and a nuclear counterstain. We have included this more explicitly in the Methods.

4. The authors used a liberal threshold of $p < 0.05$ (unadjusted) to identified 2049 transcripts that are differentially expressed. How many of these passed $p < 0.05$ (adjusted)? How would the results change when adjusted pvalue is used.

Reviewer 2 asked the same question. Here is our response:

We now include results for a series of P-values thresholds to demonstrate the robustness of our results to the threshold applied. For each threshold we ran an interaction analysis, testing the dependence on the high density of CpG sites of the relationship between normalized change in transcript abundance and the presence of mutations.

The table below summarizes the findings and is now included in the Supplemental Material in the section entitled "Sensitivity to assumptions about CpG sequence density and expression changes"

LogP threshold	Interaction effect	Interaction pval	Interaction logp
0.5	0.05	1.49E-05	4.83
1	0.06	1.87E-09	8.73
2	0.07	3.17E-12	11.5
3	0.07	8.14E-15	14.09
4	0.07	1.66E-13	12.78
5	0.07	5.98E-15	14.22
6	0.07	1.47E-13	12.83
7	0.07	4.51E-13	12.35
8	0.06	4.78E-12	11.32
9	0.07	4.78E-15	14.32
10	0.08	9.25E-17	16.03
11	0.08	7.70E-17	16.11
12	0.08	4.47E-17	16.35

5. In the supplementary, the authors looked into the effect of density in multiple cell types but didn't address the association of CpG mutations on RNA transcript abundance depending on local chromatin state.

We considered whether the effect of mutations on RNA transcript abundance might depend on local chromatin state in the following way. We used a set of chromatin state annotations of the mouse genome based on over 900 datasets from various cell and tissue types {Vu, 2023 #47229} and compared the fit of two different models to predict the difference in expression between C57BL/6J and DBA/2J. The first model included mutations, chromatin state, and density (high vs low). The second model included these elements as well as an interaction between mutation and density (i.e. allowing the effect of mutation to vary according to CpG density). We compared the two models with an analysis and found a significant improvement ($P = 8.41e-08$),

Res. Df	Res. Sum Sq	Df	Sum Sq.	F	P-value
7,215,542	5,499,959				
7,215,541	5,499,937	1	21.88	28.7	8.41E-08

We conclude that interaction between density and mutation acts independently of chromatin states.

This result is described in the main text (page 14) and in the Supplemental material. The results from running the two linear models are provided here for ease of reference:

Null model results:					
Feature	Df	Sum sq	Mean sq	F-value	Pr(>F)
States	16	4864	304.03	398.87	<2.20E-16
Density	1	2578	2578.44	3382.72	<2.20E-16
Mutation	1	119	118.51	155.48	<2.20E-16
Residuals	7215542	5499959	0.76		

Full model results:					
Feature	Df	Sum sq	Mean sq	F-value	Pr(>F)
States	16	4864	304.03	398.87	<2.20E-16
Density	1	2578	2578.44	3382.73	<2.20E-16
Mutation	1	119	118.51	155.48	<2.20E-16
Density:Mut.	1	22	21.88	28.71	8.41E-08
Residuals	7215541	5499937	0.76		

Referees' reports, second round of review

Reviewer #2: The authors have replied all of my concerns.

Reviewer #3: The authors nicely reflected the suggestions by conducting additional analysis on other clusters and clarifying the limitations of this study in the discussion section. Importantly, the authors addressed the use of nominal p-value in the analysis by providing a series of p-value thresholds, which shows the robustness of their studies.

Authors' response to the second round of review

none